# Transplantation of high-risk donor livers after resuscitation and viability assessment using a combined protocol of oxygenated hypothermic, rewarming and normothermic machine perfusion: study protocol for a prospective, single-arm study (DHOPE-COR-NMP trial)

Yvonne de Vries,[1] Tim A Berendsen,[1] Masato Fujiyoshi,[1] Aad P van den Berg,[2] Hans Blokzijl,[2] Marieke T de Boer,[1] Frans van der Heide,[2] Ruben H J de Kleine,[1] Otto B van Leeuwen,[1] Alix P M Matton,[1] Maureen J M Werner,[1] Ton Lisman,[3] Vincent E de Meijer,[4] Robert Porte[1]

For numbered affiliations see end of article.

**Correspondence to**
Dr Robert Porte;
r.j.porte@umcg.nl

## ABSTRACT

**Introduction** Extended criteria donor (ECD) livers are increasingly accepted for transplantation in an attempt to reduce the gap between the number of patients on the waiting list and the available number of donor livers. ECD livers; however, carry an increased risk of developing primary non-function (PNF), early allograft dysfunction (EAD) or post-transplant cholangiopathy. Ischaemia-reperfusion injury (IRI) plays an important role in the development of these complications. Machine perfusion reduces IRI and allows for reconditioning and subsequent evaluation of liver grafts. Single or dual hypothermic oxygenated machine perfusion (DHOPE) (4°C–12°C) decreases IRI by resuscitation of mitochondria. Controlled oxygenated rewarming (COR) may further reduce IRI by preventing sudden temperature shifts. Subsequent normothermic machine perfusion (NMP) (37°C) allows for ex situ viability assessment to facilitate the selection of ECD livers with a low risk of PNF, EAD or post-transplant cholangiopathy.

**Methods and analysis** This prospective, single-arm study is designed to resuscitate and evaluate initially nationwide declined ECD livers. End-ischaemic DHOPE will be performed for the initial mitochondrial and graft resuscitation, followed by COR of the donor liver to a normothermic temperature. Subsequently, NMP will be continued to assess viability of the liver. Transplantation into eligible recipients will proceed if all predetermined viability criteria are met within the first 150 min of NMP. To facilitate machine perfusion at different temperatures, a perfusion solution containing a haemoglobin-based oxygen carrier will be used. With this protocol, we aim to transplant extra livers. The primary endpoint is graft survival at 3 months after transplantation.

**Ethics and dissemination** This protocol was approved by the medical ethical committee of Groningen, METc2016.281 in August 2016 and registered in the Dutch Trial registration number

## Strengths and limitations of this study

► This protocol combines the benefits (resuscitation and ex situ viability testing) of multiple machine perfusion modalities (dual hypothermic oxygenated machine perfusion (DHOPE), controlled oxygenated rewarming (COR) and normothermic machine perfusion (NMP)).

► The rewarming phase (COR) links DHOPE and NMP and helps to avoid sudden temperature shifts that may cause additional injury to an already compromised donor organ.

► A newly developed perfusion fluid containing a haemoglobin-based oxygen carrier allows for perfusion at different temperatures, yet is not officially registered in the Netherlands.

► Initially, nationwide declined high-risk donor livers, which carry an increased risk of early graft failure due to primary non-function, early allograft dysfunction or post-transplant cholangiopathy, will be accepted for this machine perfusion protocol.

**Trial registration number** NTR5972, NCT02584283.

## INTRODUCTION
### Increased use of ECD livers
Driven by the donor organ shortage, extended criteria donor (ECD) or suboptimal quality livers are increasingly used for transplantation. These livers would have been refused for transplantation in the past, because of donor characteristics (advanced age, obesity, alcohol abuse and long intensive care unit (ICU) stay with use of vasopressors)

or graft pathology (steatosis, donation after circulatory death (DCD), elevated liver enzymes and history of liver trauma), but are nowadays accepted due to the shortage of suitable donor livers. ECD livers carry an increased risk of developing primary non-function (PNF), early allograft dysfunction (EAD) or post-transplant cholangiopathy.[1] These complications are difficult to predict, causing potentially transplantable ECD livers to be declined. It has been predicted that in the USA the donor liver discard rate will exceed 50% in the next two decades.[2 3] In 2016, in the Netherlands, 235 donation procedures were effectuated, yet 75 liver grafts were not used, equalling a discard rate of 32%.[4]

### Machine perfusion as a tool to resuscitate ECD livers

In previous animal and preclinical human liver studies, machine perfusion has been applied as a tool to improve and evaluate donor livers prior to transplantation.[5–7] Many machine perfusion protocols exist, with varying temperature, perfusion solution and other characteristics. Oxygenated hypothermic machine perfusion (HMP), which is performed at 4°C–12°C, aims to resuscitate the mitochondria and sustain adenosine 5'-triphosphate (ATP) production through the supply of oxygen, while the cold keeps metabolic rates suppressed.[8–10] Schlegel et al have shown that 2 hours of oxygenated HMP is sufficient to replenish cellular energy stores.[9] A pilot study by our research group, which included 30 DCD livers, has shown a 100% graft survival in the dual hypothermic oxygenated perfusion (DHOPE) group (n=10), compared with 70% in the static cold storage (SCS) group (n=20). Furthermore, the incidence of non-anastomotic strictures (NAS) of the biliary tree was 10% in the DHOPE group, compared with 35% in the SCS group.[10] Based on these favourable results, our centre initiated a multicentre randomised controlled trial on DHOPE.

Several research groups have investigated the use of mid/subnormothermic machine perfusion (NMP) or controlled oxygenated rewarming (COR), performed at 12°C–34°C.[8] COR may function as a bridge between hypo and NMP, omitting a sudden temperature shift, but may also have an ischaemia-reperfusion injury (IRI) reducing effect.[7] An abrupt temperature shift contributes to mitochondrial dysfunction, and is more pronounced during reperfusion after HMP, than after COR.[7 11] COR has also been described to increase cellular energy content in a clinical study. Furthermore, in a clinical trial performed by the group in Essen, ECD graft survival was a 100% at 6 months after transplantation in the COR group, whereas this was 85% in the SCS group.[7 11]

### Machine perfusion as a tool to evaluate ECD livers

A more accurate determination of organ viability can be provided by NMP, which aims to resume metabolism, requiring a more complex perfusion solution that includes an oxygen carrier and nutrients. Whereas NMP is a more complex method, metabolic parameters can be combined with indicators of graft injury to reflect the viability of the donor graft, giving the transplant team an indication whether the organ is suitable for transplantation.

Several research groups have proposed viability criteria to evaluate livers during NMP. Our group has suggested that 2.5 hours of NMP is sufficient to determine whether a liver is potentially transplantable.[12] Livers that produced ≥10 grams bile within 2.5 hours and ≥4 g in the preceding hour were associated with lower levels of transaminases, while glucose and lactate levels normalised. As a result, bile production was proposed as a viability criterion to identify potentially transplantable ECD livers.[12] Watson et al were the first to report on end-ischaemic NMP of suboptimal human livers. Poor prognostic factors were an inability to maintain a normal pH, slow lactate fall and an increase in alanine aminotransferase (ALT) in the perfusion fluid.[13] This group also indicated that if bile produced during NMP does not have a pH >7.45, there is a high risk of post-transplant cholangiopathy.[14] The accuracy of bile pH, bicarbonate and glucose as biomarkers of severe bile duct injury was subsequently demonstrated by our group.[15 16] Mergental et al proposed the following criteria after their first series of viability testing during NMP; perfusate lactate <2.5 mmol/L and sufficient bile production in combination with one of the following criteria; pH of perfusate >7.3, stable arterial and portal flow and homogeneous graft perfusion with soft consistency of the parenchyma.[17] The criteria for the VITTAL (Viability testing and transplantation of marginal livers) trial—a machine perfusion study using discarded livers in the UK—are similar, but do not include a minimal amount of bile production.[18] The viability criteria of the current protocol are based on above-mentioned studies and our previous experience.

### Perfusion solution with haemoglobin-based oxygen carrier

While sufficient amounts of oxygen can be dissolved in a perfusion solution at temperatures below 20°C, an oxygen carrier is necessary at higher temperatures. Red blood cell (RBC)-based perfusion solutions are, therefore, mostly used for NMP. RBCs, are, however, relatively scarce and cannot be used at hypothermic temperatures due to increased stiffness of the erythrocyte lipid membranes, which causes haemolysis. Furthermore, immune reactions can be triggered while using RBC.[19 20]

Haemoglobin-based oxygen carriers (HBOCs) are alternatives for RBC. HBOC-201, a bovine-derived haemoglobin product, has been used in both transfusion and machine perfusion studies.[21–23] Recently, HBOC-201 has received FDA approval for the application in selected US centres to treat patients with a life-threatening anaemia who are not able/willing to receive RBC.[21 22] Furthermore, several preclinical studies on NMP with an HBOC-based perfusion solution have been performed. Laing et al have shown that reactive oxygen species production, cell necrosis and apoptosis are not more pronounced in HBOC perfused livers, compared with RBC perfused livers.[19] Our group has demonstrated higher bile

## Box 1   Viewability criteria

► Cumulative bile production of ≥10 mL during first 150 min of normothermic machine perfusion (NMP) and ≥4 mL in the last hour (12).
► Lactate concentration in perfusate is (0.5–1.7 mmol/L) within 150 min of NMP (13,17).
► Perfusion fluid pH is (7.35–7.45) within 150 min of NMP, without the need for repeated addition of $NaHCO_3$ (13,17).
► Biliary pH of >7.45 within 150 min of NMP (15,29).

A liver graft will be deemed transplantable, if all of the criteria were met within 150 min of NMP. As described in the Introduction and in the Methods section, criteria were based on previous studies and own experience, and reflect both hepatocellular and biliary viability.

production and lower (ALT) release during NMP with an HBOC-based perfusion solution, compared with an RBC-based perfusion solution.[24] To facilitate machine perfusion at all temperatures and to eliminate the disadvantages of RBC usage, we have developed an HBOC-201-based perfusion solution, which will be used for the here described machine perfusion protocol.

### End-ischaemic ex situ machine perfusion protocol

In this protocol, we describe a prospective, single-arm study to resuscitate and evaluate initially nationwide declined ECD livers, using a combined protocol of DHOPE, COR and NMP. Initially, nationwide declined ECD livers are statically stored on ice and transported to our centre. DHOPE (8°C–10°C) will then be performed for 1 hour, to replenish ATP stores and diminish IRI. To facilitate a smooth transition from hypothermia to normothermia, livers will be slowly warmed up to 37°C

during an hour, using COR. Subsequently, viability testing will be performed using NMP. If all of the predetermined viability criteria are met, the liver will be transplanted into an eligible recipient. Machine perfusion will be performed using a novel HBOC-201-based perfusion solution for all temperature phases.

## METHODS AND ANALYSIS
### Study design
This study is a prospective, single-arm, single-centre study. High-risk ECD livers that are declined for liver transplantation by all Dutch liver transplant centres will be accepted for this trial. Typically, these donor livers are at increased risk of developing early graft loss due to PNF, EAD or NAS.

Organs will be allocated in compliance with Eurotransplant (ET) rules, as described below. Organs will be preserved by SCS and transported to the University Medical Center Groningen (UMCG). On arrival, liver grafts will undergo DHOPE for 1 hour, followed by 1 hour of gradual rewarming until a temperature of 37°C is reached. During 2.5 hours of NMP, a viability assessment will be carried out. The liver will be labelled as transplantable if it meets all predefined viability criteria (box 1). If a liver meets all viability criteria, the transplant procedure will proceed. If not, the liver will be offered back to ET for reallocation via the standard rules, or secondarily discarded if no other centre accepts the liver. Figure 1 shows a timeline from procurement of the donor liver until transplantation or secondary discard/secondary offer to ET for allocation.

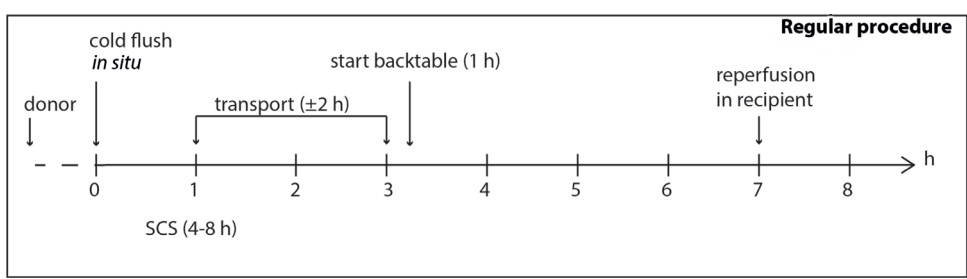

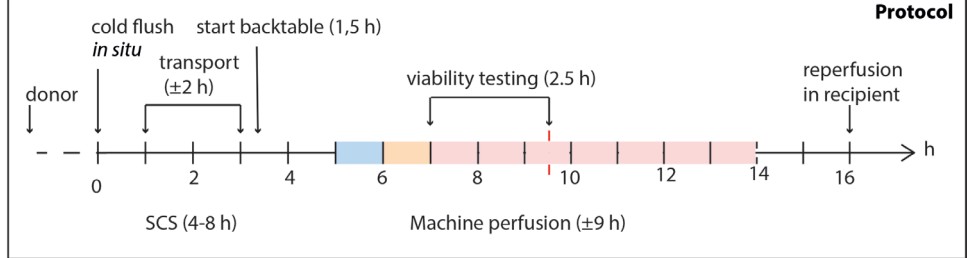

**Figure 1** Timeline of a regular liver transplantation versus the described protocol. The coloured bar depicts the machine perfusion protocol. The light blue bar represents 1 hour of DHOPE, the light orange bar represents 1 hour of COR and the pink bar represents NMP. If the liver is deemed transplantable within 150 min of NMP, NMP will continue (indicated in pink). Note that SCS and transport are approximate. Cold ischaemia time (CIT) is defined as the time from the start of cold in situ flush in the donor until reperfusion in the recipient. In the proposed protocol, CIT is defined as the time from the start of cold flush in situ until the start of machine perfusion. DHOPE, dual hypothermic oxygenated machine perfusion; NMP, normothermic machine perfusion; SCS, static cold storage.

Surgical procedure, postoperative care and follow-up are identical to our routine liver transplantation practice. Patients will be continuously monitored during their hospital stay, and subsequently at routine visits (1 month and 3 months post-transplantation).[25] After 3 months, graft survival will be evaluated to assess short-term complications; if the 3 months graft survival is less than 80%, the DHOPE-COR-NMP protocol will be considered unsuccessful. A follow-up period of 3 months after transplantation is chosen, as PNF and EAD typically occur early after transplantation.[26] Although the primary study endpoint graft survival is set at 3 months, all transplant recipients will be followed for at least 12 months to monitor for potential late complications.

### Study objective and endpoints

The study described in this protocol aims to increase the number of transplantable donor livers, by resuscitating and evaluating initially nationwide declined ECD livers through a combined protocol of DHOPE, COR and NMP, using an HBOC-based perfusion solution.

### Primary study endpoint

The primary endpoint is graft survival at 3 months after transplantation, where graft survival is defined as the absence of retransplantation or patient death. Both overall graft survival and graft survival censored for patient death (decease of the patient with a functioning graft) will be calculated.

### Secondary study endpoints

- ► Graft and patient survival at 7 days and 3, 6 and 12 months.
- ► PNF: absence of or minimal function of a liver graft after transplantation requiring retransplantation or leading to patient death within 7 days after the procedure.[27]
- ► EAD defined as one or more of the following criteria[28]: Recipient serum aspartate aminotransferase (AST) and/or ALT >2000 IU/mL within the first 7 days after transplantation.
  International normalised ratio (INR) of ≥1.6 on postoperative day 7.
  Recipient serum bilirubin levels of ≥171 µmol/L (10 mg/dL) on postoperative day 7.

- ► Development of clinically evident NAS (biliary strictures in presence of symptoms such as elevated cholestatic enzymes, jaundice, cholangitis or pruritus). The number of interventions, such as endoscopic retrograde cholangiopancreatography, percutaneous transhepatic cholangiography and drain and reoperation, will also be recorded.
- ► Biochemical analysis of graft function and IRI determined by postoperative serum levels of ALT, AST, alkaline phosphatase, gamma-glutamyl transferase, INR, lactate dehydrogenase, creatinine, platelets and total bilirubin at postoperative day 0–7 and 1 and 3 months.

### Donor and recipient inclusion and exclusion criteria

Adult patients (≥18 years old) on the UMCG waiting list for a liver transplantation, who gave informed consent for this study, will be included. Inclusion and exclusion criteria are summarised in table 1. Eligible patients will be asked for informed consent during the screening process for liver transplantation by a hepatologist or researcher who is well known with the topic and assigned to this task by the delegation log.

Donor risk factors precluding safe transplantation, such as concurrent donor malignancy and a fibrotic/cirrhotic macroscopic appearance of the liver, are considered exclusion criteria for the study. Furthermore, an agonal phase of >2 hours in DCD donors is considered an exclusion criteria, as it is national policy for procurement teams to withdraw if circulatory arrest has not occurred within 2 hours after withdrawal of life support. However, formally long donor warm ischaemia time is not an exclusion criterion for this study.

### Study procedures
#### Allocation procedure

After the liver of a donor in the Netherlands is declined by all three centres for regular transplantation following SCS alone, it will again be offered to our centre for inclusion in this protocol. In compliance with ET rules, livers will be allocated only to patients who are identified in the ET match list, based on the ET allocation rules. If a liver is not accepted for a patient who has given informed consent for the DHOPE-COR-NMP trial,

| Table 1 | Inclusion and exclusion criteria | |
| --- | --- | --- |
| | **Inclusion** | **Exclusion** |
| Donor | Bodyweight ≥40 kg | HIV, hepatitis B or C positive<br>Split or partial liver grafts<br>Domino donor livers<br>Expected CIT ≥10 hours (20) |
| Recipient | ≥18 years old<br>Informed consent | HIV positive<br>Mental incapacitation<br>Fulminant liver failure or retransplantation for PNF<br>Participation in another trial which might influence outcomes of this trial |

CIT, cold ischaemia time; PNF, primary non-function.

**Table 2** Composition of the HBOC-201-based perfusion solution

| Component | Manufacturer/distributor | Volume (mL) |
|---|---|---|
| HBOC-201 (Hemopure) | HbO$_2$ therapeutics, Pennsylvania, USA | 1250 |
| Gelofusine 4% | B Braun, Melsungen, Germany | 300 |
| Albumin 20% | Sanquin (Dutch blood bank) | 250 |
| Total parenteral nutrition (N14G30E) | Hospital pharmacy | 20 |
| Addamel (trace elements) | Fresenius Kabi, the Netherlands | 10 |
| Metronidazol (Flagyl) 5 mg/mL | Baxter BV, Utrecht, the Netherlands | 44 |
| Sterile water | B Braun, Melsungen, Germany | 335 |
| Insulin (NovoRapid) 100 IU/mL | Novo Nordisk BV, Alphen aan den Rijn, the Netherlands | 1 |
| Cernevit (multi vitamins) | Baxter BV, Utrecht, the Netherlands | 2 |
| Heparin (5000 IU/mL) | Leo Pharma, Amsterdam, the Netherlands | 2 |
| Cefazolin 1 g/5 mL | Baxter BV, Utrecht, the Netherlands | 2 |
| Taurocholic acid sodium salt 0.1% (1 mg/mL) | Sigma Aldrich, Saint Louis, USA | 7.7 |
| Sodium bicarbonate 8.4% | B Braun, Melsungen, Germany | 35 |
| KCl 1 mmol/mL | B Braun, Melsungen, Germany | 2 |
| Glutathion (Tationil) 600 mg/4 mL | Teofarma, Pavia, Italy | 14 |
| Total volume | | 2274.7 |

HBOC, haemoglobin-based oxygen carrier.

it will be allocated to another centre in one of the ET member states, according to ET allocation rules. ET and the national competent authority, the Dutch Transplantation Foundation (Nederlandse Transplantatie Stichting) have agreed that this protocol does not interfere with the regular allocation rules within the Netherlands or ET.

### Preparation of the liver

The donor liver will be procured by one of the national multiorgan procurement teams. A standard surgical technique of in situ cold flush via the aorta with 4–7 L of University of Wisconsin (UW) cold storage solution (0°C–4°C), supplemented with 50 000 IU of heparin, will be used. If possible, the liver will be procured with a segment of 3–5 cm supratruncal aorta left attached to the coeliac trunk. The portal vein and common bile duct will be kept as long as possible. After procurement, the liver will be flushed via the portal vein with at least 1 L of UW cold storage solution. The cystic duct will be ligated, and the bile duct will be gently flushed with Belzer UW cold storage solution. The liver will be kept on ice (SCS) during transport to the UMCG where cannulation of the supratruncal aorta, portal vein, and bile duct will be performed. Furthermore, a small catheter will be inserted into the caval vein to collect venous perfusate samples.

### Preparation of the perfusion solution

The perfusion solution containing HBOC-201 (table 2) and taurocholic acid for continuous infusion will be prepared under sterile conditions. The perfusion fluid was developed by our research group and has been used successfully in a preclinical study and the first clinical patients.[24 25] For the purpose of this study, the perfusion fluid was evaluated and tested for stability and compatibility by the pharmacy of the UMCG. Sterile preparation of the perfusion fluid will be conducted by the pharmacy of the UMCG. Taurocholic acid will be bought from Sigma Aldrich (Saint Louis, USA) and subsequently prepared for clinical use by pharmacy A 15 (the Netherlands), according to good manufacturing practice (GMP). Taurocholic acid will be continuously infused into the perfusion solution at a rate of 7.7 mg/hour from the start of the NMP phase. Furthermore, sodium bicarbonate can be added to the perfusion solution to correct a low pH.

### Perfusion device

The Liver Assist (Organ Assist, Groningen, the Netherlands) is a CE marked (European Union Certification of Safety, Health and Environmental Requirements) machine perfusion device for ex situ perfusion of donor livers. The Liver Assist enables perfusion of the liver via the portal vein and hepatic artery, using two centrifugal pumps to provide continuous and pulsatile flow, respectively. The system is pressure controlled, which provides auto regulation of the flow through the liver. The temperature can be set from 8°C to 37°C, and the preservation solution can be oxygenated by two hollow fibre membrane oxygenators.

### Machine perfusion settings, viability assessment and subsequent transplantation

The Liver Assist will be primed with the HBOC-201-based perfusion solution. The machine perfusion protocol consists of 1 hour of DHOPE, followed by 1 hour of COR, and a minimum of 2.5 hours of NMP. During DHOPE, the fraction of inspired oxygen (FiO$_2$) will be set at a 100% and the O$_2$ flow at 1 L, as described previously.[10] During

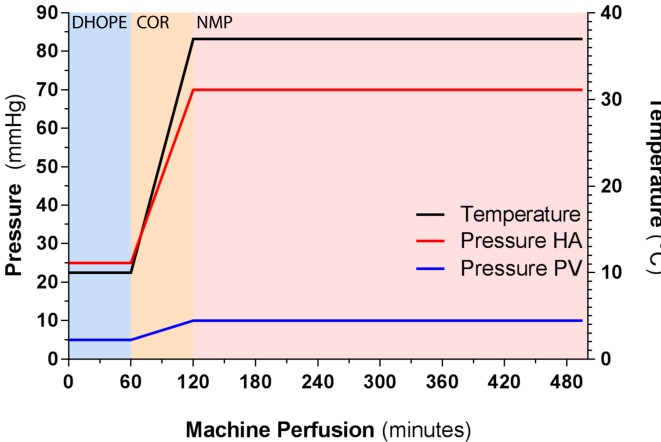

**Figure 2** Liver Assist pressure and temperature settings during DHOPE, COR and NMP. COR, controlled oxygenated rewarming; DHOPE, dual hypothermic oxygenated perfusion; HA, hepatic artery; NMP, normothermic machine perfusion; PV, portal vein.

COR and NMP, the $FiO_2$ and $O_2$ flow will be adjusted according to arterial partial pressure of oxygen ($pO_2$) and venous saturation at the level of the suprahepatic inferior vena cava, where arterial $pO_2$ should be 10.0–13.3 kPa and venous saturation 55%–75%. During COR, the temperature and arterial and portal pressure will be gradually increased, as depicted in figure 2.

During the first 2.5 hours of NMP, viability assessment will be carried out. The liver should match all of the criteria as described in box 1. These criteria were derived from literature and our own experience and reflect both hepatocellular and biliary viability.[12 13 15 17 29] If the liver meets the predefined viability criteria, the recipient operation will be started. NMP of the liver will continue until recipient hepatectomy is nearly complete. The liver will then be disconnected from the machine and immediately flushed with 2 L cold Belzer UW cold storage solution to remove the HBOC-201-based perfusion fluid. If the liver does not meet the predefined viability criteria, the liver will be offered back to ET for re allocation or secondary discard.

### Follow-up

After transplantation, patients will be monitored and treated according to standard post-transplant care. After discharge from the hospital, patients will be evaluated in the outpatient clinic up to 3 months. A summary of outcome parameters is shown in box 2.

---

**Box 2    Laboratory investigations during follow-up**

► Aspartate aminotransferase.
► Alanine aminotransferase.
► Alkaline phosphatase.
► Gamma-glutamyl transferase total bilirubin.
► Lactate dehydrogenase.
► Creatinine.
► Platelets.
► International normalised ratio.

---

### Statistics

A total of 10 liver transplantations will be included in this study to determine if the proposed protocol can be applied safely. To transplant 10 livers, we expect to assess 20 nationwide declined donor livers, as our preclinical study has indicated a potential recovery rate of at least 50%.[30] The intervention group will not be compared with a control group, as the livers that will be used are initially nationwide declined livers and it would not be ethical to transplant these livers directly. It is our main concern that the livers are safely transplanted and can match postoperative outcomes of regularly transplanted livers. Graft survival at 3 months after transplantation was chosen as the primary endpoint, as described in the ' Methods section'. An acceptable graft survival would be at least >80%, based on data of regular transplantations.[31] If the 3-month graft survival rate is <80%, the trial will be stopped. Primary and secondary outcomes are described under 'Study objective and endpoints'.

### Collection, storage and analyses of data and samples
#### During machine perfusion and liver transplantation

► Perfusion characteristics, such as flow through the portal vein and hepatic artery, pressure settings for the portal vein and hepatic artery, resistance in the portal vein and hepatic artery, and temperature, will be recorded every 15 min.
► Every half hour arterial perfusate, and every hour venous perfusate, will be sampled for point of care analyses to evaluate liver function. Perfusate samples will be stored at −80°C for later laboratory analyses, such as the assessment of hepatocellular injury markers and function.
► From COR/NMP onwards bile will be collected and stored at −80°C for later assessment of biliary injury. Furthermore, every half hour bile samples will be collected under mineral oil for point of care analysis to determine cholangiocyte function.
► Biopsies of liver parenchyma and common bile duct will be taken prior to machine perfusion, after machine perfusion and after reperfusion in the recipient. Biopsies will be snap frozen or stored in formalin for paraffin embedding. H&E staining of paraffin biopsies will be performed to analyse hepatobiliary histological injury.
► Haemodynamic status (mean arterial pressure, heart rate and the use of inotropic medication) of the recipient will be recorded prior to, during and after graft reperfusion.

### Other study parameters: baseline values and parameters at inclusion

► Patient general demographics (age, gender, weight and height).
► Patient medical history, including:
► Model for end-stage liver disease score.
► Indication for transplantation.

de Vries Y, et al. BMJ Open 2019;**9**:e028596. doi:10.1136/bmjopen-2018-028596

- ► Symptoms of decompensated cirrhosis (ie, ascites, encephalopathy and hepatorenal syndrome).
- ► Viral status (ie, hepatitis A–E; cytomegalovirus (CMV); Epstein-Barr virus (EBV))
- ► Current hospitalisation status (home, hospital ward or ICU).
- ► Kidney function, as assessed by creatinine, glomerular filtration rate and the need for dialysis.
- ► Medication.
- ► Donor and liver graft characteristics (age, gender, weight before and after machine perfusion, height, cause of death, viral status, liver function, amount of steatosis and donor risk index).
- ► Reason for initial decline of the liver graft.
- ► In case of DCD, characteristics such as time interval between withdrawal of life support and circulatory arrest, time interval between circulatory arrest and start cold perfusion in situ, donor hepatectomy time.
- ► Surgical methods and technical difficulties or abnormalities.
- ► Cold ischaemia time, total preservation time (including SCS, DHOPE and NMP) and warm ischaemia time during implantation.

## Monitoring and reporting

Serious adverse events will be reported to the accredited medical ethical committee through the national web portal ToetsingOnline within 15 days after detection. Monitor visits via the Trial Coordination Centre (official institute for the monitoring of clinical trials in our centre) will take place after inclusion of 1, 5 and 10 patients. During these visits, the monitor will examine the line listing of all adverse events.

## Patient and public involvement

Patients and/or public were not involved in the design of the study and the selection of outcome parameters. No patients are involved in the recruitment and/or conduct of the study. Dissemination of the results of the study will be via scientific congresses and meetings, publication(s) in peer-reviewed scientific journal(s), and may include a press release for lay media. Participants in the trial will be informed about the outcomes through their local investigators.

## DISCUSSION

ECD livers are increasingly accepted for transplantation, due to the persistent shortage of suitable donor livers, leading to waiting list morbidity and mortality. Although ECD livers are increasingly used, many livers are still declined for transplantation because the estimated risk of early graft loss after transplantation, such as PNF, EAD or biliary complications, is considered too high.[1] The current study aims to increase the number of livers suitable for transplantation by resuscitating and evaluating initially nationwide declined ECD livers with a combined machine perfusion protocol, using an HBOC-based perfusion solution.

The risks associated with participation in this study are related to the use of ECD livers. However, current literature and data from our own research group suggest that the proposed protocol will be beneficial for such livers, leading to acceptable outcome after transplantation. DHOPE has been described to resuscitate mitochondria, increase ATP concentration and reduce IRI of both the hepatocytes and cholangiocytes.[8–10] Subsequently, COR prevents (mitochondrial) damage during and after a sudden temperature shift. Lastly, NMP will be used to assess viability.[7 11]

The primary endpoint of this study is based on the increased risk of ECD livers to develop PNF, EAD and post-transplant cholangiopathy.[1] As these complications mostly occur within 3 months, graft survival at 3 months after transplantation was chosen as the primary outcome.[26] If graft survival is >80%, we will consider the proposed protocol as feasible and safe. A graft survival of <80% will be considered a cut-off point for this trial. These percentages are based on outcome of regular liver transplantations.[31]

One of the down sides of using high-risk ECD liver that were initially nationwide declined is the inability to add a control group. In our opinion, it would be unethical to transplant livers with a perceived high risk of complications, without resuscitating and evaluating these livers with machine perfusion prior to transplantation. For studies that include initially declined donor livers it is not unusual to not include a control group, as the risk of developing severe complications would be too high. The VITTAL trial has a similar design.[18]

To combine the machine perfusion phases, DHOPE, COR and NMP, an HBOC-201-based perfusion solution, will be used. In vivo side effects of first generation, HBOCs have been observed in the past. However, HBOC-201 recently received FDA approval for application in selected centres to treat patients with life-threatening anaemia, who are unable/unwilling to receive RBC.[21 22] The use of HBOC-201 for machine perfusion is an ex situ application and three different groups have reported favourable preclinical results after machine perfusion with an HBOC-201-based perfusion solution.[19 23 24] To mitigate the potential side effects of HBOC-201 liver grafts will be flushed thoroughly with regular Belzer SCS solution after the machine perfusion procedure.

In summary, with this protocol of sequential DHOPE, COR and NMP, we aim to resuscitate and select initially declined suboptimal donor livers for transplantation, in order to increase the number of transplantable livers. The application of the combined machine perfusion protocol will be considered safe and feasible if graft survival at 3 months after transplantation of these initially declined donor livers is similar to that of regularly transplanted donor livers.

## ETHICS AND DISSEMINATION

As described in the Methods section, the perfusion solution will be prepared by our hospital pharmacy according

to GMP, to ensure sterility and composition of the product. Monitoring of this study will be carried out by the trial coordination centre of the UMCG after inclusion of 1, 5 and 10 transplantations to ensure compliance with the protocol and patient safety in this trial.

Patient data will be coded and can only be decrypted via a secured document. After 15 years, data will be destroyed. All team members involved with this trial will adhere to the code of good research conduct. The results of this study will be submitted to a peer-reviewed international medical journal. Data of this study might be used for side studies.

**Author affiliations**
[1]Hepatobiliary Surgery and Liver Transplantation, University of Groningen, University Medical Center Groningen, Groningen, The Netherlands
[2]Gasteroenterology and Hepatology, Universitair Medisch Centrum Groningen, Groningen, The Netherlands
[3]Surgical Research Laboratory, University of Groningen, University Medical Center Groningen, Groningen, The Netherlands
[4]Surgery, Universitair Medisch Centrum Groningen, Groningen, The Netherlands

**Contributors** The study was initiated and designed by YdV and RP. YdV, MF, APvdB, HB, MTdB, FvdH, RHJdK, OBvL, APMM, MJMW, TL, VEdM and RP contributed to the final study design, which was determined during several meetings. The manuscript was drafted by YdV, TAB and RP. YdV, TAB, MF, APvdB, HB, MTdB, FvdH, RHJdK, OBvL, APMM, MJMW, TL, VEdM and RP edited the manuscript and approved the final version of the manuscript.

**Funding** This work is supported by the Dutch Ministry of Health Welfare and Sport and the de Cock—Hadders Foundation, Groningen. The HBOC-201 is kindly provided by HBO2 Therapeutics (Souderton, USA).

**Competing interests** None declared.

**Patient consent for publication** Obtained.

**Ethics approval** This study is investigator initiated and was approved by the medical ethical committee of Groningen, METc2016.281 in August 2016. Furthermore, approval was given by the Dutch Transplantation Society and the Eurotransplant organisation to use initially nationwide declined donor livers for the proposed protocol. The device used for this study is CE marked, and its use was approved by the competent authority as well.

**Provenance and peer review** Not commissioned; externally peer reviewed.

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
