## [Reviewer comments · BMJ Open]

ARTICLE DETAILS

TITLE (PROVISIONAL)	Transplantation of high-risk donor livers after resuscitation and viability assessment using a combined protocol of oxygenated hypothermic, rewarming and normothermic machine perfusion: Study protocol for a prospective, single arm study (DHOPE – COR – NMP Trial)
AUTHORS	de Vries, Yvonne; Berendsen, Tim; Fujiyoshi, Masato; van den Berg, Aad; Blokzijl, Hans; de Boer, Marieke; van der Heide, Frans; de Kleine, Ruben; van Leeuwen, Otto; Matton, Alix; Werner, Maureen; Lisman, Ton; De Meijer, Vincent; Porte, Robert

VERSION 1 – REVIEW

REVIEWER	Mr Thamara Perera Consultant Transplant Surgeon The Liver Unit, Queen Elizabeth Hospital Birmingham and Birmingham Women's & Children's Hospital Birmingham United Kingdom
REVIEW RETURNED	14-Jan-2019

GENERAL COMMENTS	The study protocol submitted for BMJ Open to be considered for publication under the theme of "Transplantation of high-risk donor livers after resuscitation and viability assessment using a combined protocol of oxygenated hypothermic, rewarming and normothermic machine perfusion: a prospective, single arm study: DHOPE – COR – NMP Trial" by the research group led by Porte et al was reviewed with great interest. Machine perfusion has been embraced by transplant community at a rapid pace over the last few years, with many options range from hypothermic to normothermic temperatures, and short term resuscitation to continuous preservation and current focus is on selecting the optimal approach. Whilst it is well known that a good proportion of quality liver grafts do not need any type of intervention, as it is unlikely to change the course of post operative outcomes in terms of graft and transplant related post operative recovery the benefits of machine perfusion are deemed to be more for the marginal or so called extended criteria grafts (ECD). A sizeable proportion of ECD grafts are currently go unutilised. These grafts, if proven to be functional outside a human body, may help the clinician confidence on utilisation thereby benefitting a number of patients on transplant wait lists. In this background the proposed approach has a substantial ground for testing the hypothesis that initial mitochondrial resuscitation followed by controlled oxygenated rewarming and final viability testing at
--

	normothermic temperatures helps to increased utilisation of ECD liver grafts with outcome measures of graft survival at 90-days. the study protocol is well written and concise, easy to understand apart from one section under the statistics which I cannot fully agree to. Authors expect the 90-day survival to be over 80%, a threshold which is too low for standard liver transplant procedures, as many transplant programs nowadays achieve over 95% patient and graft survival, however for extended criteria grafts this may be an acceptable figure. Nevertheless, their aim is to transplant 10 liver grafts according to this protocol and what would you do if the first two transplants do not succeed? your protocol states that the trial would be stopped, and I do not fully agree with this approach. The threshold for accepting or declining a liver graft relies of the size of the transplant program, expertise of the surgeons and the burden on the transplant wait list. For example, a particular graft deemed marginal by a smaller transplant centre may be deemed acceptable another centre. Declining organs based on the donor details/visual examination by a "procurement surgeon" alone may not be accurate to classify an organ as ECD, and many organs that are classed as ECD are not be ECD in real sense. This is a bias that is very common in any of the studies of this nature, and there is no clear way of overcoming this. Please explain how do you MINIMISE this bias? For example, a liver initially declined by other centres in the country may be offered to your centres, the quality of the graft may be such that in your experience it could have been used. On the other hand when you have this trial up and running there may be occasions where you might think that it is safer to decline the liver on SCS so that it could enter the trial. Another concern that I have with regards to the clarity of figure 1 outlining study pathway, which I think is difficult to comprehend, and moreover gives that idea that authors are considering to the organs with short cold ischaemic time to be enrolled to the study, contrary to the inclusion criteria (CIT>10hours) depicted in table 2.
--	--

REVIEWER	Ina Jochmans Abdominal Transplantation KU Leuven University Hospitals Leuven Belgium
REVIEW RETURNED	15-Jan-2019

GENERAL COMMENTS	This is the study protocol of an ongoing study looking at feasibility and safety of human liver viability testing before transplantation. An innovative protocol of HMP-COR-NMP with HBOC-201 as an oxygen carrier is used. I have some questions regarding the submitted protocol that I hope can be addressed as they might help the readership understand the set-up of the study in more detail:  - You recently published the outcome of the first 5 livers that were included in this trial (NTR5972) in the American Journal of Transplantation (Am J Transplant. 2018 Dec 26. doi: 10.1111/ajt.15228). This points toward the fact that intermittent analysis of outcomes were performed, though there is no mention of planned intermittent
--

	analyses in the protocol. Was this analysis planned? If so, this should be mentioned in the protocol. If not, why was it performed? Also, and perhaps not common in publication of trial protocols, I believe it would be valuable to refer to this publication in the trial protocol as it gives additional relevant information to the readership. - The study includes nationwide declined ECD livers, i.e. livers that were declined for transplantation by all Dutch centres. From the description in the protocol, I understand that these livers are being offered for this study protocol before they are offered outside of the Netherlands, as would be Eurotransplant protocol at the stage of 'rescue allocation'. Although regular allocation of the liver takes place on a national level, a deviation from regular allocation will take place either as 'extended allocation' or 'rescue allocation' when livers prove difficult to place within regular allocation – such as one might expect of ECD livers (Chapter 5 of the Eurotransplant manual, www.eurotransplant.org). At the stage of rescue allocation, the liver will be offered to the centres in the region or country where the liver is located at the time. Should the liver not be accepted for transplantation by these centres, centres in a wider geographical range are contacted and – as a last possibility – centres outside Eurotransplant are contacted. Where in the process of liver allocation is offering for this study protocol considered by Eurotransplant? Were Eurotransplant allocation rules changed for this study? Is there not a possibility for inherent selection bias as risk-appetite can vary substantially between centres and countries? In other words, might livers that are included in this protocol not be accepted and transplanted elsewhere without viability assessment? Would an (international) comparator group of such livers not be informative? - Regarding inclusion and exclusion criteria for the donor and donor liver: You mention that HBV and HCV positive livers are excluded. Is there a particular reason for this as such livers do provide an accepted source of liver grafts for some patients (e.g. HCV positive recipients)? You state that "high-risk ECD livers that were declined for liver transplantation by all Dutch liver transplant centres will be accepted for this trial". However, I cannot imagine that all declined livers are accepted in reality (i.e. donor cancer, anatomical injuries, etc.). Are there other criteria besides the ones mentioned that are taken into account to decide whether or not a liver that is offered or the trial will be accepted? - From Figure 1, I understand that the potential recipient is called in before the decision to transplant is made. Is it correct that they are called in at time the liver is accepted into the protocol? When are patients approached to take part in this study? Do you consider all liver transplant candidates as potential recipients for these high-risk livers or do you preferentially allocate them to the sickest liver transplant candidates? As these livers will be offered to you as a centre-offer, which criteria are used to pick a patient for a particular liver that is going to be assessed as I can imagine there might be more than 1 suitable patient on your waiting list?
--	---

	- Regarding endpoints How is graft loss defined in this study? Is patient survival at 3 months not a secondary endpoint? What are clinically evident non-anastomotic biliary strictures? Need for diagnostic procedures, need for treatment? Will vascular resistances be calculated? - Regarding the perfusate Am I correct to say that there are no continuous infusions that are given to the liver during machine perfusion? Bicarbonate is only part of the perfusion solution but is not given when pH falls, correct? The pharmacy tested stability of the perfusion solution. How long is it stable? Can it be prepared in advance and 'shelved', or does it need preparation on demand? Is the UMCG pharmacy GMP certified? - Regarding oxygen delivery: you aim for a 'venous saturation' of 55-75%, you mean at the outflow of the hepatic veins? - A total of 10 liver transplants will be included and the study will be stopped when 3-month graft survival rates are < 80%. Could you please clarify what you mean by this? That means that interim analysis will be performed (confer my comment earlier), how else would you be able to determine 3 month graft survival rates? After how many cases were these planned? Will the study stop after the inclusion of 10 livers? And if not, how many will be included? - Regarding Ethics: you mention patient data will be stored anonymously with decrypting details available. Would that not mean data are pseudoanonymised as data can be retraced to the recipient via a (secured) file? This would be an important difference to make with regard to applicability of GDPR regulation. - Figure 1, a minor detail: multi organ donation is used, however, I can imagine that a liver from a liver-only donor would also be eligible. - Figure 2 depicts the changes in temperature and pressure settings. The temperature axis is not very detailed (ranging from 0°C to 100°C), during COR what are the incremental steps of temperature increase?
--	---

REVIEWER	Vasilis Kosmoliaptsis MD, PhD, FRCS Department of Surgery, University of Cambridge, UK
REVIEW RETURNED	24-Jan-2019

GENERAL COMMENTS	I read with great interest the study protocol submitted by de Vries and colleagues. The authors intend to resuscitate primarily declined extended criteria donor (ECD) livers by a combination of end-ischemic dual hypothermic oxygenated perfusion (DHOPE - 4-12°C) followed by controlled oxygenated rewarming (COR) and subsequently normothermic machine perfusion (NMP - 37°). Livers will be assessed during NMP and if predetermined viability criteria are fulfilled the livers will be transplanted into eligible recipients.
---

	The authors aim to include 10 patients and the study has started in mid-2017 and will be completed in 2019. The study protocol is unique in that it combines different currently clinically available machine perfusion types, DHOPE, COR and NMP, in one setting. This can only be achieved because first the authors use a perfusion machine which allows adjustment of the perfusion temperature, and second, the authors use a perfusate with a haemoglobin-based oxygen carrier (HBOC) allowing the change of temperature. Additionally, they have a well-established donation after circulatory (DCD)-program and are experienced with using ECD grafts. The study protocol is well written and the aims are clearly described. One limitation of the study is that it will not answer the question which perfusion modality is superior and whether all three steps (DHOPE, COR and NMP) are necessary. There are some minor comments the authors should address prior to publication:  • The authors aimed to use high-risk ECD livers. They should describe in more detail which types of ECD will be included and what is the max. WIT for DCD donors? • Graft survival at 3-months has been chosen as the primary endpoint. This is a short observation period and the authors argue that this is the time frame when most of the complications related to transplantation of ECD grafts occur (Primary non-function, early allograft dysfunction and post-transplant cholangiopathy – all secondary endpoints). However, as cited in the introduction, graft survival of 100% at 6 months has already been reported by the Essen group, transplanting ECD grafts after COR. Therefore, graft survival at 6 months would be a more relevant endpoint. • Please describe how non-anastomotic strictures will be diagnosed? Will the patients receive a routine MRCP after transplantation? And at what time points will those be performed? • % of patients on the waiting list with informed consent?
--	--

REVIEWER	Peter Friend University of Oxford, UK Involved in trials of normothermic liver perfusion. Cofounder and Chief Medical Officer of spin-out company involved in commercialisation of organ perfusion - not as used in this protocol
REVIEW RETURNED	26-Jan-2019

GENERAL COMMENTS	This is an interesting and novel trial proposal, combining for the first time the techniques of hypothermic machine perfusion, controlled oxygenated reperfusion and normothermic machine perfusion, using a novel oxygen carrier to replace red blood cells. The study is based on high quality preclinical data and comes from an group with excellent credentials in this field. My only suggestion relates to the statistical design - although this is a reasonable size for what is a Phase-1 safety and feasibility study, it would be interesting to know how the trial size of 10 was arrived at? Also, the stopping rules could be described more clearly - 80% survival at 3 months is relatively crude in a safety study - will there be an interim analysis after a certain number of patients have been treated? If available, this information would be of value
---

VERSION 1 – AUTHOR RESPONSE

Reviewer: 1

1. Authors expect the 90-day survival to be over 80%, a threshold which is too low for standard liver transplant procedures, as many transplant programs nowadays achieve over 95% patient and graft survival, however for extended criteria grafts this may be an acceptable figure. Nevertheless, their aim is to transplant 10 liver grafts according to this protocol and what would you do if the first two transplants do not succeed? your protocol states that the trial would be stopped, and I do not fully agree with this approach.

We agree with reviewer 1 that an 80% survival threshold for a typical, benchmark liver transplantation (i.e., heart-beating young donor, primary full size transplantation without steatosis) is too low. The Dutch Transplantation Foundation (Nederlandse Transplantatie Stichting) has reported on a 1-year graft survival of 75 %, a 6-month graft survival of 80% and a 3-month graft survival of slightly over 80 % in recipients receiving either a DBD liver or a donation after circulatory death (DCD) liver. [<https://www.transplantatiestichting.nl/cijfers/hoe-lang-gaat-een-donororgaan-mee/hoe-lang-gaat-een-donorlever-mee>]. In addition, international studies have shown a 1-year graft survival of approximately 90% after 'regular' DCD liver transplantation [Schlegel et al. J Hepatol 2019]. Furthermore, it has been described that an increasing DRI score is associated with decreasing survival rates [Braat et al. Am J Transplant 2012]. Based on these numbers a cut off value of 80% graft survival at three months after transplantation seems acceptable for this study, in which we will include nationwide initially declined extended criteria (ECD) livers (with a high DRI). If 3-month graft survival is less than 80%, the DHOPE-COR-NMP protocol will be considered unsuccessful, because this would indicate that this machine protocol is not safe.

2. The threshold for accepting or declining a liver graft relies of the size of the transplant program, expertise of the surgeons and the burden on the transplant wait list. For example, a particular graft deemed marginal by a smaller transplant centre may be deemed acceptable another centre. Declining organs based on the donor details/visual examination by a "procurement surgeon" alone may not be accurate to classify an organ as ECD, and many organs that are classed as ECD are not be ECD in real sense. This is a bias that is very common in any of the studies of this nature, and there is no clear way of overcoming this. Please explain how do you MINIMIZE this bias? For example, a liver initially declined by other centres in the country may be offered to your centres, the quality of the graft may be such that in your experience it could have been used. On the other hand when you have this trial up and running there may be occasions where you might think that it is safer to decline the liver on SCS so that it could enter the trial.

We understand the concern of the reviewer. On page 9 under the section 'Study design' we have stated the following : 'Typically, these donor livers are at increased risk of developing early graft loss due to PNF, EAD or NAS.' This implies that all donor livers that will be included in this study are at increased risk of developing one of these complications. Livers that are more prone to develop PNF, EAD or NAS have one or a combination of the following risk factors ; high donor age; donation after circulatory death (and inherent warm ischemia time); elevated liver enzymes; high BMI, long ICU stay and alcohol abuse [Merion et al. J Hepatol 2016; Mirza et al. Lancet 1994]. Livers that will be included in this study have more than one of the previously mentioned risk factors. These risk factors will be collected, as stated in the manuscript in the subsection 'Collection, storage and analyses of data and samples' on page 15.

Furthermore, we would like to emphasize that all livers have to be declined for regular transplantation by all three Dutch transplant centres (including our own). This is described in the subsection 'Study

design' on page 9 and under 'Allocation procedure' at Page 12. Using this as a selection criterion, we have tried to minimize the risk of selection bias.

3. Another concern that I have with regards to the clarity of figure 1 outlining study pathway, which I think is difficult to comprehend, and moreover gives that idea that authors are considering to the organs with short cold ischaemic time to be enrolled to the study, contrary to the inclusion criteria (CIT>10hours) depicted in table 2.

We acknowledge the comment from the reviewer and have adjusted this figure accordingly.

Reviewer: 2

1. You recently published the outcome of the first 5 livers that were included in this trial (NTR5972) in the American Journal of Transplantation (Am J Transplant. 2018 Dec 26. doi: 10.1111/ajt.15228). This points toward the fact that intermittent analysis of outcomes were performed, though there is no mention of planned intermittent analyses in the protocol. Was this analysis planned? If so, this should be mentioned in the protocol. If not, why was it performed? Also, and perhaps not common in publication of trial protocols, I believe it would be valuable to refer to this publication in the trial protocol as it gives additional relevant information to the readership.

This single arm, open label trial will be analyzed continuously to ensure safe implementation of this new technique. There will be no formal, planned interim analysis. All (serious) adverse events, however, will be actively reported to a data safety monitoring committee. Although the primary study endpoint is graft survival at 3 months, all transplant recipients will be followed for at least 12 months to monitor for potential late complications. We have added this to the subsection 'Study design' on page 9. Here we have also included a reference to our publication of the first five patients enrolled in this trial. The primary purpose of this publication was primarily to report on the safety and feasibility of machine perfusion with an acellular hemoglobin-based oxygen carrier-containing perfusion solution. Therefore, we have also included this reference in the paragraph on 'Preparation of the perfusion solution' on page 13 of the revised manuscript.

2. The study includes nationwide declined ECD livers, i.e. livers that were declined for transplantation by all Dutch centres. From the description in the protocol, I understand that these livers are being offered for this study protocol before they are offered outside of the Netherlands, as would be Eurotransplant protocol at the stage of 'rescue allocation'. Although regular allocation of the liver takes place on a national level, a deviation from regular allocation will take place either as 'extended allocation' or 'rescue allocation' when livers prove difficult to place within regular allocation – such as one might expect of ECD livers (Chapter 5 of the Eurotransplant manual, www.eurotransplant.org). At the stage of rescue allocation, the liver will be offered to the centres in the region or country where the liver is located at the time. Should the liver not be accepted for transplantation by these centres, centres in a wider geographical range are contacted and – as a last possibility – centres outside Eurotransplant are contacted. Where in the process of liver allocation is offering for this study protocol considered by Eurotransplant? Were Eurotransplant allocation rules changed for this study?

Eurotransplant (ET) and the national competent authority, the The Dutch Transplantation Foundation (Nederlandse Transplantatie Stichting) were contacted during the design of this protocol and they agreed that the study design does not interfere with the regular allocation rules within the Netherlands or ET. Dutch livers are always first offered to Dutch centres for regular transplantation, unless there is a recipient with an HU (high urgency) or ACO (approved combined organ) status in one of the other ET member states. If a high-risk donor liver is not accepted for regular transplantation by all three liver transplant centers in the Netherlands, it will again be offered to our center with the question if we have

a potential recipient on our waiting list who appears in the regular ET matching list and who has given informed consent for a DHOPE-COR-NMP preserved liver. If so, the liver will be allocated to that patient. If there is no matching recipient, the liver will be allocated to a patient outside our country, following the regular ET rules. The ET Liver and Intestine Transplant Advisory Committee (ELIAC) has agreed with this protocol and concluded that it does not deviate from any of the current allocation rules within ET.

In the revised manuscript we have added abovementioned information on page 11 and 12.

Furthermore we have added to the subsection 'Study design' on page 9 and the subsection 'Machine perfusion settings, viability assessment and subsequent transplantation' on page 13, that the donor liver will again be offered to ET if it does not meet the viability criteria.

3. Is there not a possibility for inherent selection bias as risk-appetite can vary substantially between centres and countries? In other words, might livers that are included in this protocol not be accepted and transplanted elsewhere without viability assessment? Would an (international) comparator group of such livers not be informative?

We kindly refer to our response on comment 2 of reviewer 1. It would indeed be informative to perform an (international) comparator study. This could be considered for a successive study.

4. Regarding inclusion and exclusion criteria for the donor and donor liver: You mention that HBV and HCV positive livers are excluded. Is there a particular reason for this as such livers do provide an accepted source of liver grafts for some patients (e.g. HCV positive recipients)?

If we would accept HBV or HCV positive livers we would run the risk of contaminating our liver perfusion device. Obviously, disposables will be renewed for every case, but the device itself not. This is in analogy to dialysis machines used for renal replacement therapy, where the use of dedicated devices for HBV and HCV positive patients is common practice.

5. You state that "high-risk ECD livers that were declined for liver transplantation by all Dutch liver transplant centres will be accepted for this trial". However, I cannot imagine that all declined livers are accepted in reality (i.e. donor cancer, anatomical injuries, etc.). Are there other criteria besides the ones mentioned that are taken into account to decide whether or not a liver that is offered or the trial will be accepted?

We acknowledge the comment from the reviewer that not all declined livers will be considered for inclusion in the study. Only livers that are nationwide declined because of an estimated too high risk of early graft loss (i.e. due to primary non-function, severe early allograft dysfunction and/or severe biliary complications), will be eligible for inclusion in this trial. We have described this better in the subsection 'Donor and recipient in- and exclusion criteria' of the revised manuscript (Page 11).

6. From Figure 1, I understand that the potential recipient is called in before the decision to transplant is made. Is it correct that they are called in at time the liver is accepted into the protocol?

That is correct.

7. When are patients approached to take part in this study? Do you consider all liver transplant candidates as potential recipients for these high-risk livers or do you preferentially allocate them to the sickest liver transplant candidates?

Eligible patients will be asked for informed consent during the screening process for liver transplantation/before being put on the waiting list. All potential recipients are considered, except the ones that match an exclusion criteria. New candidates for liver transplantation are discussed during our weekly multidisciplinary liver transplantation meeting. For each patient it will be decided whether that patient is a candidate for participation in this study. We have added this information to the subsection 'Donor and recipient in- and exclusion criteria' on page 11.

A liver is allocated to the highest matching patient on the waiting list, who gave informed consent for participation in the DHOPE-COR-NMP study. We have added this to the subsection 'allocation procedure' on page 11.

8. As these livers will be offered to you as a centre-offer, which criteria are used to pick a patient for a particular liver that is going to be assessed as I can imagine there might be more than 1 suitable patient on your waiting list?

According to the Eurotransplant rules, livers are not offered as a center-offer. Livers are only offered to individual patients who appear on the Eurotransplant matching list. As described under comment 7, we can accept a liver (after nationwide decline for regular transplantation) for the highest matching patient on our waiting list who gave informed consent for this study.

9. How is graft loss defined in this study?

Graft loss is defined as either retransplantation or patient death. Both graft survival and graft survival censored for patient death (decease of the patient with a functioning graft) will be calculated. We have added this to the section 'Study objective and endpoints' on page 10.

10. Is patient survival at 3 months not a secondary endpoint?

We acknowledge that patient survival at 3 months is a secondary endpoint and have added this to the subsection 'Secondary study endpoints' on page 10.

11. What are clinically evident non-anastomotic biliary strictures? Need for diagnostic procedures, need for treatment?

Clinically evident non-anastomotic biliary strictures (NAS) are NAS that are diagnosed in the presence of biliary symptoms such as elevated cholestatic enzymes, jaundice, cholangitis or pruritus. Usually, this type of clinical symptoms are an indication for imaging of the biliary tree. Routine imaging will not be performed in this study. We have added this to the subsection 'Secondary study endpoints' on page 11.

12. Will vascular resistances be calculated?

Yes, vascular resistances will be calculated. We have added this to the protocol on page 15 in the subsection 'Collection, storage and analyses of data and samples'

13. Am I correct to say that there are no continuous infusions that are given to the liver during machine perfusion?

During NMP continuous infusion of taurocholate into the perfusion solution will take place. We have added this to subsection 'Preparation of the perfusion solution' on page 12.

14. Bicarbonate is only part of the perfusion solution but is not given when pH falls, correct?

This is partly correct. Bicarbonate is both part of the baseline perfusion solution, but boluses can be given when pH falls during NMP. We have added this to the subsection 'Preparation of the perfusion solution' on page 12. In table 1 ' Viability criteria' of the submitted manuscript is stated: Perfusion fluid pH is [7.35 – 7.45] within 150 min of NMP, without the need for repeated addition of NaHCO₃. This implies that bicarbonate may be administered to correct perfusate acidosis. Amounts of extra bicarbonate added during machine perfusion will be recorded.

15. The pharmacy tested stability of the perfusion solution. How long is it stable? Can it be prepared in advance and 'shelved', or does it need preparation on demand? Is the UMCG pharmacy GMP certified?

The perfusion solution is stable for one week. The stock of perfusion solution is therefore refreshed every week, or earlier whenever necessary after use. We confirm that the UMCG pharmacy is GMP certified.

16. Regarding oxygen delivery: you aim for a 'venous saturation' of 55-75%, you mean at the outflow of the hepatic veins?

We apologize for the confusion and confirm that we meant the 'venous at the outflow level of the hepatic veins. To avoid this confusion, we have changed 'venous saturation' to 'venous saturation at the level of the suprahepatic inferior vena cava' in the subsection ' Machine perfusion settings, viability assessment and subsequent transplantation' on page 13.

17. A total of 10 liver transplants will be included and the study will be stopped when 3-month graft survival rates are < 80%. Could you please clarify what you mean by this? That means that interim analysis will be performed (confer my comment earlier), how else would you be able to determine 3 month graft survival rates? After how many cases were these planned? Will the study stop after the inclusion of 10 livers? And if not, how many will be included?

We kindly refer to our response on comment 1. In this study we will include 10 liver transplantations after which an analysis will be performed to assess the aforementioned study outcomes. In case of satisfactory graft- and patient survival, a successive study including more livers will be initiated. If we fail to obtain a graft survival rate of at least 80%, we will consider this method as unsuccessful and discontinue its application in the current format.

18. Regarding Ethics: you mention patient data will be stored anonymously with decrypting details available. Would that not mean data are pseudoanonymised as data can be retraced to the recipient via a (secured) file? This would be an important difference to make with regard to applicability of GDPR regulation.

Patient data is coded and can only be used via a secured file that is accessible by the sponsor and trial coordinator. We have added this to the section 'Ethics and dissemination' one page 20.

19. Figure 1, a minor detail: multi organ donation is used, however, I can imagine that a liver from a liver-only donor would also be eligible.

That is correct and we have therefore adjusted Figure 1.

20. Figure 2 depicts the changes in temperature and pressure settings. The temperature axis is not very detailed (ranging from 0°C to 100°C), during COR what are the incremental steps of temperature increase?

During COR the temperature will be gradually increased with about 1°C per 2 minutes. We have adjusted the temperature axis of Figure 2.

Reviewer: 3

1. The study will include donor livers that are deemed unsuitable for transplantation (declined) by all Dutch liver transplant centres. It would be useful to include specific graft inclusion/exclusion criteria to enable more meaningful analysis of the study's outcomes, comparison with similar studies in this field, and assessment of the specific reasons for liver decline. It is not clear how the marginal condition of the donor liver is defined (e.g. donor risk index cut off, degree of liver steatosis, warm ischaemia for DCD donor livers etc.)

A similar question was raised by Reviewer 1, item 2. We, therefore, kindly refer the reviewer to our response to comment 2 by Reviewer 1. Unfortunately, there are no universally accepted criteria for a non-transplantable liver. Whether to accept a liver for transplantation depends on a weighing of risks and this may vary between centres and surgeons within a centre. In the current protocol we have therefore included the rule that a liver has to be declined (for medical reasons) by all three transplant centers in the Netherlands. This does not exclude that a liver would not be considered acceptable by any other center in Europe, but at least the three Dutch centres have independently from each considered the liver 'unacceptable for regular transplantation'.

2. Similarly, the liver recipient inclusion criteria are very broad (any adult patient with appropriate consent) but it would be worth clarifying this point; are the investigators going to include high-risk recipients e.g. any retransplant patient (I note the exclusion of re-transplant patients for PNF) or patients with portal vein thrombosis or high MELD score etc. (if high-risk recipients are excluded then a definition of the criteria and how this will be done is required).

We do include all recipients that match the inclusion criteria and give informed consent. Patients requiring a retransplantation and patients with portal vein thrombosis are also eligible for inclusion in this study.

3. A justification of the statistical aspects of the study design would be desirable. The main aim of the study is safety of the proposed intervention although the investigators also state that the study: 'aims to increase the number of livers suitable for transplantation...'. Why a total of 10 liver transplants? Is there any provision and justification for the number of donor livers that the study is going to perfuse/assess (donor liver sample size)? If the feasibility of the intervention to rescue discarded livers is an objective of the study then a justification of the donor liver sample size that will be included in the study and the anticipated conversion rate (number of transplants divided by number of perfused livers) merits clarification (for example, are the investigators prepared to assess 100 livers to transplant 10 or is there a target donor liver sample size?). Is there a hypothesis for any desirable liver recovery rate?

We agree that the sample size requires additional clarification. With a primary study end point of 3-month graft survival to indicate safety and feasibility, a total number of 10 livers was determined and approved by the medical ethical committee.

With regards to the liver recovery rate, our previous pre-clinical study indicates a potential recovery rate of at least 50% (Westerkamp AC et al. Transplantation 2016). We therefore expect to assess 20

nationwide discarded donor livers to allow 10 successful liver transplantations. We have added this to the section statistics on page 14.

4. It is important to clarify if any interim analysis of the study's primary endpoint (graft survival of at least 80% at 3 months post-transplant) is going to be performed. Is this going to be assessed after a total of 10 transplants are performed? What if the first few cases lead to graft failure? Will the investigators assess the futility of the intervention at an early stage and if so, how (e.g. using a staged design)? What is the role of the data monitoring committee in any interim analysis?

This single arm, open label trial will be analyzed continuously to ensure safe implementation of this new technique. There will be no formal, planned interim analysis. All (serious) adverse events, however, will be actively reported to a data safety monitoring committee. Although the primary study endpoint is graft survival at 3 months, all transplant recipients will be followed for at least 12 months to monitor for potential late complications.

5. What is the follow up period for documentation of adverse effects (e.g. beyond 3 months)? Are there any contraindications to the use of HBOC-201?

Although the primary study endpoint is graft survival at 3 months, all transplant recipients will be followed for at least 12 months to monitor late complications.

There are no contraindications for the use of HBOC-201. Several studies have described safe application of HBOC-201 infusion in humans [Mer et al. Transfusion 2016; Davis et al. Transfusion 2018]. Furthermore, HBOC-201 based perfusion solutions for ex situ machine perfusion have previously been used, with favorable results [Matton et al. Liver Transplant 2018; Fontes et al. Am J Transplant, 2015; Laing et al. Transplantation 2017]. We would like to emphasize that the HBOC-201-containing perfusion solution is flushed out of the liver after machine perfusion, and that only a small fraction of HBOC-201 is expected to enter the circulation of the recipient.

6. Please define 'clinically evident non anastomotic strictures'. Is cholangiography going to be performed in all cases?

We kindly refer to our response on comment 11 of reviewer 2.

7. DCD: please correct definition to 'donation after circulatory death' (page 6, line 19).

We thank you for pointing this out. We have changed this (on page 5).

8. Page 8, line 30: bile production is included in the VITTAL study

We acknowledge this comment and have rephrased this on page 7 of the revised manuscript.

9. Figure 2: the right hand side axis does not seem to depict temperature?

It does, however the scale is quite large. We have therefore adjusted the temperature axis of Figure 2.

10. The investigators might want to consider including a reference to the largest RCT in this field (Nasralla et. Nature 2018; 557: 50).

We acknowledge the interesting study by Nasralla and coworkers on normothermic machine preservation in humans. Although this indeed is the largest randomized controlled trial in the field of human liver machine perfusion, many fundamental differences between Nasralla's study and our protocol exist. The most essential technical difference is that in the Nasralla trial (normothermic) machine preservation was applied during transport, whereas in our study all livers will be transported on ice (traditional 'static cold storage'). In other words, Nasralla et al. have applied NMP as an alternative preservation method for static cold storage, whereas we have applied NMP after a period of static cold storage. Moreover, the trial by Nasralla et al. included 'regular' donor livers which were considered acceptable for transplantation, whereas in our study we only include high-risk extended criteria donor livers, which are nationwide declined for transplantation.

Reviewer: 4

1. The authors aimed to use high-risk ECD livers. They should describe in more detail which types of ECD will be included and what is the max. WIT for DCD donors?

We kindly refer to our response to comment 2 of reviewer 1. In the national DCD protocol in the Netherlands, a maximum time period between withdrawal of life support and circulatory arrest is 2 hours. The procurement team will withdraw if a potential DCD donor has not arrested within 2 hours. Other than that there is no formal limitation to donor warm ischemia time. We have added this to the section 'Donor and recipient in- and exclusion criteria' on page 11.

2. Graft survival at 3-months has been chosen as the primary endpoint. This is a short observation period and the authors argue that this is the time frame when most of the complications related to transplantation of ECD grafts occur (Primary non-function, early allograft dysfunction and post-transplant cholangiopathy – all secondary endpoints). However, as cited in the introduction, graft survival of 100% at 6 months has already been reported by the Essen group, transplanting ECD grafts after COR. Therefore, graft survival at 6 months would be a more relevant endpoint.

We acknowledge the comment of the reviewer. High-risk donor livers are at higher risk of developing primary non-function (PNF), early allograft dysfunction (EAD) and post-transplant cholangiopathy. PNF and EAD by definition occur within the first two weeks after transplantation. Post-transplant cholangiopathy, especially due to ischemia induced bile duct injury, mostly occurs within three months after transplantation [Den Dulk et al. Transplant Direct 2015; Buis et al. Liver Transplant 2007]. We therefore choose to set our primary end-point at 3-month graft survival, to assess feasibility and safety of this protocol. However, all patients will be followed for at least 12 months to monitor long-term outcomes and potential long-term adverse effects. We have described this better in the subsection 'Study design' on page 9.

3. Please describe how non-anastomotic strictures will be diagnosed? Will the patients receive a routine MRCP after transplantation? And at what time points will those be performed?

Clinically evident non-anastomotic biliary strictures (NAS) are NAS that are diagnosed in the presence of biliary symptoms such as elevated cholestatic enzymes, jaundice, cholangitis or pruritus. Usually, this type of clinical symptoms are an indication for imaging of the biliary tree. Routine imaging will not be performed in this study. We have added this to the subsection 'Secondary study endpoints' on page 11.

4. % of patients on the waiting list with informed consent?

We expect that about 75% of adult patients will provide informed consent for this study.

Reviewer: 5

My only suggestion relates to the statistical design - although this is a reasonable size for what is a Phase-1 safety and feasibility study, it would be interesting to know how the trial size of 10 was arrived at? Also, the stopping rules could be described more clearly - 80% survival at 3 months is relatively crude in a safety study - will there be an interim analysis after a certain number of patients have been treated? If available, this information would be of value

We thank you for the suggestion. We kindly refer to our response on comment 1 of reviewer 1, response to comment 1 of reviewer 2 and response to comment 3 of reviewer 3.

VERSION 2 – REVIEW

REVIEWER	Mr. Thamara Perera The Liver Unit, Queen Elizabeth Hospital Birmingham United Kingdom
REVIEW RETURNED	01-Apr-2019
GENERAL COMMENTS	Authors have satisfactorily answered my queries and a congratulations of the putting together this protocol
REVIEWER	Ina Jochmans University Hospitals Leuven, Belgium
REVIEW RETURNED	03-Apr-2019
GENERAL COMMENTS	Thank you for addressing my questions.
REVIEWER	Vasilis Kosmoliaptsis MD, PhD, FRCS Department of Surgery, University of Cambridge, UK
REVIEW RETURNED	08-Apr-2019
GENERAL COMMENTS	The authors have addressed the points raised in their revised manuscript.
REVIEWER	Dagmar Kollmann, MD, PhD Department of Surgery, Medical University of Vienna, Austria
REVIEW RETURNED	14-Apr-2019
GENERAL COMMENTS	The authors responded adequately to the reviewers requests and the manuscript is clear and well-written. Two minor remarks: Page 14: the last sentence is not completed: primary and secondary outcomes are described under 'Study objective and ... Page 20: revise the first sentence of the second paragraph: Patient data will >< coded and ...
REVIEWER	Peter Friend University of Oxford, UK Academic and commercial interests in normothermic organ perfusion: : Co-founder of OrganOx Ltd

	: PI on clinical trials of alternative machine perfusion technology
REVIEW RETURNED	06-Apr-2019

GENERAL COMMENTS	The authors have responded to the comments made. I believe that the omission of MRCPs as a means of diagnosing the incidence of biliary strictures is a defect in the study design that may result in the results being more difficult to publish in due course. Clinical manifestation of biliary obstruction is a very crude measure, particularly in the context of such a small trial. The manufacturer of taurocholate should be identified. 'Hospital Pharmacy' is presumably the location where the medication is formulated rather than synthesised; in the context of a medical product used presumably outside its licence (or possibly unlicensed) this is important.
--

VERSION 2 – AUTHOR RESPONSE

Reviewer: 4

Two minor remarks:

Page 14: the last sentence is not completed: primary and secondary outcomes are described under 'Study objective and ...

Page 20: revise the first sentence of the second paragraph: Patient data will >< coded and ...

Response: Thank you for making us aware of this. We have adjusted these sentences.

Reviewer: 5

I believe that the omission of MRCPs as a means of diagnosing the incidence of biliary strictures is a defect in the study design that may result in the results being more difficult to publish in due course. Clinical manifestation of biliary obstruction is a very crude measure, particularly in the context of such a small trial.

Response: We appreciate the reviewer's opinion. For this single-arm study, we indeed choose to monitor clinical manifestations of biliary strictures instead of performing MRCPs routinely, as we found this clinically most relevant. Imaging of the biliary tree will of course be performed when clinically indicated.. Laboratory results as well as results of any imaging studies of the biliary tree will be recorded to assess the number of clinically relevant biliary strictures. We have added this to the manuscript on page 11.

The manufacturer of taurocholate should be identified. 'Hospital Pharmacy' is presumably the location where the medication is formulated rather than synthesised; in the context of a medical product used presumably outside its licence (or possibly unlicensed) this is important.

Response: The hospital prepares the taurocholate for application, however it does not manufacture it. Another pharmacy called 'pharmacy A15' in the Netherlands manufactures the tauracholate for clinical use. The pharmacy A15 has a GMP certificate (good manufacturing practice) and manufactures medicines and pharmaceuticals products for our hospital pharmacy. The raw material for the taurocholate is bought from Sigma Aldrich (Saint Louis, USA). We have added this information to the manuscript on page 13 and in table 3.